# UOEP: User-Oriented Exploration Policy for Enhancing Long-Term User Experiences in Recommender Systems

## Abstract

Reinforcement learning (RL) has gained traction for enhancing user long-term experiences in recommender systems by effectively exploring users' interests. However, modern recommender systems exhibit distinct user behavioral patterns among tens of millions of items, which increases the difficulty of exploration. For example, user behaviors with different activity levels require varying intensity of exploration, while previous studies often overlook this aspect and apply a uniform exploration strategy to all users, which ultimately hurts user experiences in the long run. To address these challenges, we propose User-Oriented Exploration Policy (UOEP), a novel approach facilitating fine-grained exploration among user groups. We first construct a distributional critic which allows policy optimization under varying quantile levels of cumulative reward feedbacks from users, representing user groups with varying activity levels. Guided by this critic, we devise a population of distinct actors aimed at effective and fine-grained exploration within its respective user group. To simultaneously enhance diversity and stability during the exploration process, we further introduce a population-level diversity regularization term and a supervision module. Experimental results on public recommendation datasets demonstrate that our approach outperforms all other baselines in terms of long-term performance, validating its user-oriented exploration effectiveness. Meanwhile, further analyses reveal our approach's benefits of improved performance for low-activity users as well as increased fairness among users.

## 1 Introduction

Recommender Systems (RS) are integral components embedded within an array of web services, encapsulating e-commerce (Pei et al., 2019; Huzhang et al., 2021), social media (Wu et al., 2020b; 2019), and news feeding (Wu et al., 2020a; 2021), among others. In recent years, there has been a notable surge in the interest in Reinforcement Learning (RL) given its distinctive ability to explore user interests and elevate long-term performance within RS. In contrast to traditional supervised methods (Liu et al., 2009; Xu et al., 2018) that optimize immediate user feedback, RL-based RS (Zhao et al., 2018; Chen et al., 2019; 2023) treats the recommendation as an interactive process, focusing on maximizing the cumulative rewards of users in the long term. These RL-based methods yield a more comprehensive understanding of user preferences, thereby equipping RS to generate tailored recommendations that are congruent with user satisfaction in the long run.

Unlike other RL-based tasks, applying RL to RS in practical scenarios suffers from large action space and sparse user feedback (Dulac-Arnold et al., 2015; Liu et al., 2023). These factors contribute to intricate user behavioral patterns. For instance, most users and items exhibit only a limited number of interactions, whereas a minority of highly active users or popular items contribute to a small fraction of the total. This difference in user behavior results in a long-tailed return distribution, which in turn adds difficulties to the task of learning user preferences. Therefore, an effective RL-based recommender is expected to learn diverse policies that exploit more for the active users while exploring more for the inactive ones. To better illustrate the differences among users, we conduct experiments with the quantile-based grouping of users on a public short video dataset named KuaiRand. Specifically, we arrange users by their Click Through Rate (CTR). Subsequently, we select users from various bottom quantiles ($0 \sim \alpha$) of this sorted outcome, which represents users

with the bottom $\alpha$ activity levels. We then evaluate the performance of two typical RL algorithms under different levels of noise on these user groups, where the larger noise represents more intense exploration. Figure 7 demonstrates that as the quantile decreases, indicating users with lower activity levels, there is a tendency for them to exhibit a preference for higher levels of noise. The observation suggests that the optimal exploration intensity varies significantly across different user groups, emphasizing the importance of implementing more effective and fine-grained exploration in RL-based recommender systems.

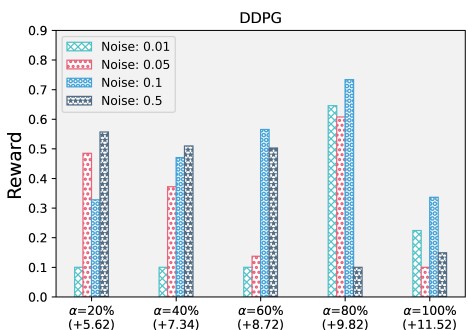 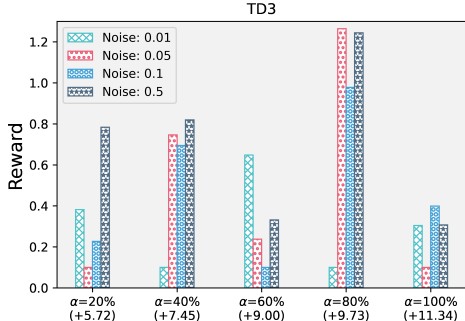

Figure 1: Illustration Experiment. We sorted users based on their activity levels (i.e., CTR) and selected the five bottom $\alpha$-quantile ($\alpha \in \{0.2, 0.4, 0.6, 0.8, 1.0\}$) of user groups. We then trained two RL algorithms, DDPG and TD3, on these five groups under four varying noise levels. We conducted all experiments with four different seeds and reported and averaged the results. For better presentation, we performed a shift on the values within each group, ensuring that the minimum value within each group becomes 0.1. Adding the value inside the parentheses on the horizontal axis yields the actual return.

Despite its critical significance, the effective exploration strategies at the user level remain largely uninvestigated. Traditional reinforcement learning exploration strategies, such as noise perturbation and $\epsilon$-greedy, introduce randomness mainly during the selection of actions (Yu, 2018). Recent studies have made strides in this area by employing decomposition techniques that make RL more manageable with recommendation slates, or by using alignment methods to balance the trade-off between exploration and exploitation in the latent action representation space (Ie et al., 2019; Liu et al., 2023; Dulac-Arnold et al., 2015). However, these methods have primarily concentrated on the regularization of action space, overlooking return distribution variations among users and thus not effectively enabling user-oriented exploration.

Based on the above observations, in this paper, we propose a novel approach called UOEP for reinforcing user-oriented exploration in recommender systems. UOEP employs the return distribution under different quantiles to explicitly characterize the activity level of users and group them according to the quantile. In this way, multiple actors can be learned and each actor corresponds to a specific user group with a predefined level of activity. Specifically, we first introduce a distributional critic to learn the return distribution instead of an expected return. Given the return distribution, multiple actors are learned where each actor subsequently optimizes towards different target quantiles within the return distribution. This approach enables us to customize the exploration intensity for different groups. Then, a population diversity regularization term is introduced to encourage diversity among the different actors, promoting effective exploration in the action space. Considering that introducing multiple actors at once may cause the instability of the training, we also incorporate a supervision module to ensure the stability during the actor learning process. We evaluate the performance of our approach and compare it with state-of-the-art recommendation approaches through comprehensive experiments on various public and industrial recommendation datasets. The experimental results demonstrate its advantages in terms of long-term performance. Further analyses reveal our model's capability to better serve low-active users while enhancing overall fairness. These supplementary benefits further highlight the superiority of our approach.

## 2 PROBLEM FORMULATION

We focus on the task of session-based recommendation in this paper. Formally, given an item candidate pool $\mathcal{I}$, the recommender system aims to select a list of items $x_t^u = \{i_1, \ldots, i_n\}$ for the current user $u$ at each time step $t$. Here, $i_k \in \mathcal{I}$ for $1 \leq k \leq n$, and $n$ represents the list size, which denotes the number of items provided to the user during each interaction within the session. The goal of the learning is to encourage users to engage in more interactions within a session (*depth*) and maximize their cumulative rewards, such as clicks or purchases over the sessions (*return*). We formulate this problem as the Markov Decision Process (MDP), which consists of 5-tuple $\langle \mathcal{S}, \mathcal{A}, \mathcal{P}, \mathcal{R}, \gamma \rangle$:

- $\mathcal{S}$ is the *continuous state space*, where $s \in \mathcal{S}$ indicates the state of a user including static features such as gender and dynamic features such as historical interactions.

- $\mathcal{A}$ is the *action space*, where $a \in \mathcal{A}$ is possible recommendation lists in a single interaction within a session, i.e., $\mathcal{A} = \mathcal{I}^n$.

- $\mathcal{P} : \mathcal{S} \times \mathcal{A} \times \mathcal{S} \rightarrow \mathbb{R}$ is the *state transition function*, where $p(s'|s, a)$ specifies the probability of transitioning from the current state $s$ to a new state $s'$ after taking action $a$.

- $\mathcal{R} : \mathcal{S} \times \mathcal{A} \rightarrow \mathbb{R}$ is the *reward function* that maps a user state $s$ and an action $a$ to an immediate reward $r(s, a)$, which is related to the user feedback, such as clicks or likes.

- $\gamma$ is the *discount factor* for weighting future rewards relative to immediate rewards.

The policy function $\pi(a|s) : \mathcal{S} \rightarrow \mathcal{A}$ represents the probability of selecting action $a$ given state $s$. We define $\mathcal{Z}^\pi(s, a)$ to represent the return distribution under $\pi$ given state $s$ and action $a$, equaling in distribution $\sum_{t=0}^{\infty} \gamma^t r(s_t, a_t)$. The objective of standard RL is to learn an optimal policy $\pi^*(a|s)$ that maximizes the expectation of $\mathcal{Z}^\pi$, denoted as $\max_{\pi^*} \mathcal{J}_{\mathbb{E}}(\pi^*) := \mathbb{E}[\mathcal{Z}^\pi]$. However, as previously stated, due to the complex distribution of user feedbacks in RS, solely maximizing expectations results in ineffective and coarse-grained exploration. In this paper, we turn to facilitate the exploration by improving the capture of the return distribution, as opposed to focusing solely on the expectation. This can be represented as $\max_{\pi^*} \mathcal{J}_{\mathbb{D}}(\pi^*) := \mathbb{D}[\mathcal{Z}^\pi]$. where $\mathbb{D}$ is a distortion operator that maps the reward distribution to real numbers. Since our goal is to improve the effectiveness of exploration under different user groups, a better choice of the distortion operator $\mathbb{D}$ is $\text{CVaR}_\alpha$, which captures the expected return when experiencing a given bottom $\alpha$-quantile of the possible outcomes (Tang et al., 2019). Mathematically, it can be represented as follows:

$$\text{CVaR}_\alpha(\mathcal{Z}^\pi) = \frac{1}{\alpha} \int_0^\alpha F^{-1}(\mathcal{Z}^\pi; \tau) d\tau, \tag{1}$$

where $\alpha \in [0, 1]$ and the *quantile function*, denoted as $F^{-1}(\mathcal{Z}^\pi; \cdot)$, is the inverse function of the cumulative distribution function (CDF) of the return distribution $\mathcal{Z}^\pi$. It maps a quantile value $\tau \in [0, 1]$ to the corresponding value of the random variable within the distribution. When $\alpha = 1$, the CVaR value is equal to the expectation of the entire distribution.

## 3 UOEP: THE PROPOSED APPROACH

In this section, we introduce our approach: User-Oriented Exploration Policy (UOEP). As depicted in Figure 2, UOEP primarily comprises a population of actors and a distributional critic. Each actor within this population operates independently to offer distinct recommendations to users. Additionally, each actor is assigned a unique $\alpha$ value, which is utilized for optimization targeting distinct bottom $\alpha$-quantiles of the return distribution, denoted as $\text{CVaR}_\alpha$. This approach ensures that each actor learns within user groups characterized by different activity levels. To enable effective exploration among actors in the population and ensure stability during the learning process, we introduce two regularization terms. We first incorporate a population diversity regularization loss $\mathcal{L}_{\text{div}}$ to facilitate effective exploration through diversified actors. Then, we introduce a supervision module $\mathcal{L}_{\text{sta}}$ to stabilize the learning process. We delve into the learning process for the distributional critic in Section 3.1. Next, in Section 3.2, we define the actor loss, which relies on a more effective distortion operator, CVaR, applied to the acquired return distribution and optimize it using a gradient-based approach. Finally, we introduce the two regularization terms, $\mathcal{L}_{\text{div}}$ and $\mathcal{L}_{\text{sta}}$ in Section 3.3 .

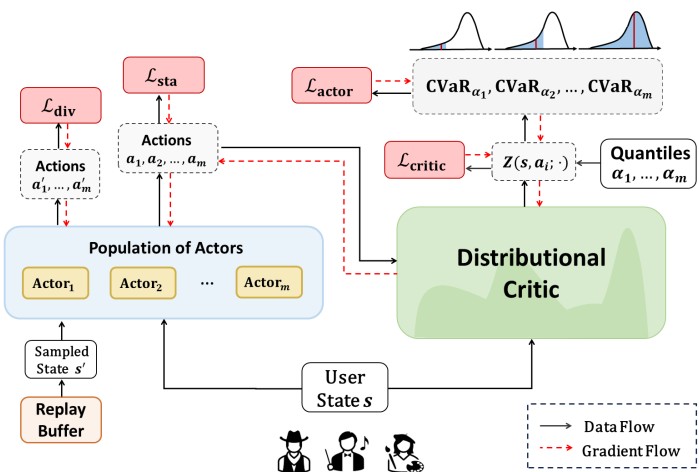

Figure 2: The proposed approach UOEP. It includes a population of $m$ actors, where $m$ is the population size and each actor$_i$ outputs an action $a_i$ based on the current user state $s$. The action $a_i$ along with state $s$ is fed into the distributional critic. Afterward, utilizing both its quantile value $\alpha_i$ and the critic's output $Z(s, a_i; \cdot)$, actor$_i$ computes the conditional value at risk (CVaR) measure in order to derive its policy gradients.

## 3.1 LEARNING DISTRIBUTIONAL CRITIC

To learn a distributional critic, we first introduce the *distributional Bellman equation*:

$$\mathcal{Z}^\pi(s, a) \overset{\mathcal{D}}{=} r(s, a) + \gamma \mathcal{Z}^\pi(s', a'), \tag{2}$$

where random variables $s', a'$ are drawn according to $s' \sim p(\cdot|s, a)$ and $a' \sim \pi(\cdot|s')$, and $A \overset{\mathcal{D}}{=} B$ denotes the equality in probability distribution between the random variables $A$ and $B$ (Bellemare et al., 2017). In practice, we choose to represent the return distribution by learning its implicit quantile function using an Implicit Quantile Network (IQN) (Dabney et al., 2018a; Urpí et al., 2021). This approach offers computational advantages for efficiently calculating CVaR. Specifically, we learn the distributional critic $Z_\theta(s, a; \tau)$ parameterized by $\theta$, to approximate the *quantile function* $F^{-1}(\mathcal{Z}^\pi(s, a); \cdot)$ at a given quantile $\tau \in [0, 1]$. Formally, for each sampled transition $(s, a, r, s')$, the temporal difference (TD) error can be computed as follows:

$$\delta_{\tau, \tau'} = r + \gamma Z'_{\theta'}(s', a'; \tau') - Z_\theta(s, a; \tau), \tag{3}$$

where $a' \sim \pi(\cdot|s')$, $\tau, \tau'$ are independently sampled from the uniform distribution, i.e., $\tau, \tau' \sim \mathbb{U}(0, 1)$ and $Z'$ is a target network whose parameters are soft-updated to match the corresponding models (Fujimoto et al., 2018). To accurately learn the relationships between quantiles, we utilize the $\tau$-quantile Huber Loss as proposed by Dabney et al. (2018b). The loss function is defined as:

$$\mathcal{L}_\kappa(\delta; \tau) = \underbrace{\left| \tau - \mathbb{1}_{\{\delta < 0\}} \right|}_{\text{quantile loss}} \cdot \underbrace{\begin{cases} \delta^2/2\kappa & \text{if } |\delta| \leq \kappa, \\ |\delta| - \kappa/2 & \text{otherwise,} \end{cases}}_{\text{Huber loss}} \tag{4}$$

where $\kappa > 0$ is a hyperparameter that controls the growth rate of the loss, and it is typically set to 1. We approximate the critic loss for all levels $\tau$ by sampling $N$ independent quantiles $\tau$ and $N'$ independent target quantiles $\tau'$, resulting in the following expression:

$$\mathcal{L}_{\text{critic}}(\theta) = \mathbb{E}_{(s, a, r, s', a') \sim \pi} \left[ \frac{1}{N \cdot N'} \sum_{i=1}^{N} \sum_{j=1}^{N'} \mathcal{L}_\kappa \left( \delta_{\tau_i, \tau'_j}; \tau_i \right) \right]. \tag{5}$$

## 3.2 OPTIMIZING POPULATION OF POLICIES TOWARDS CVAR

Now, we can start optimizing the population of actors. Building upon our previous observation, we aim to have each actor learn towards different $\tau$-quantile return distributions, which can be

achieved by computing $\text{CVaR}_\alpha$. Given a reliable distributional critic, the CVaR can be efficiently approximated using a sampling-based scheme from its quantile representation. Similar as Eq. 1, for each state-action pair $(s, a)$, we can use the critic's output $Z_\theta(s, a; \tau)$ to compute $\text{CVaR}_\alpha$ by Monte Carlo sampling:

$$\text{CVaR}_\alpha\left(\mathcal{Z}^\pi(s, a)\right) = \frac{1}{\alpha} \int_0^\alpha Z_\theta(s, a; \tau)\mathrm{d}\tau \approx \frac{1}{K} \sum_{k=1}^K Z_\theta\left(s, a; \tau_k\right), \quad \tau_k \sim \mathbb{U}(0, \alpha), \qquad (6)$$

where $K$ is the sampling times. Hence, considering an actor $\pi_{\phi_i}(a|s)$ parameterized by $\phi_i$ along with its assigned quantile $\alpha_i$, the actor loss can be written as:

$$\mathcal{L}_{\text{actor}}(\phi_i) = -\mathbb{E}_{(s,a) \sim \pi_{\phi_i}}\left[\text{CVaR}_{\alpha_i}\left(\mathcal{Z}^\pi(s, a)\right)\right], \quad i \in \{1, \ldots, m\}, \qquad (7)$$

where $m$ is the population size. Under the guidance of Eq. 7, each actor within the population learns for a specific quantile value, ensuring fine-grained exploration of user groups with varying activity levels. Furthermore, during the training process, we observed that directly setting a low value for $\alpha$ makes it difficult for the trained actors to perform well. This phenomenon is known as the *blindness to success* problem (Greenberg et al., 2022), which refers to the tendency to overlook successful cases and get stuck in local optima. To mitigate this issue, we have incorporated a soft mechanism into our actors. Instead of using a fixed level, the actor will optimize the $\alpha'$ value gradually decreasing from 1 to $\alpha$ during training. With this setting, we can compute $\alpha_t$ during training using:

$$\alpha_t = \max\left\{\alpha, 1 - \beta \cdot (1 - \alpha) \cdot t\right\}, \qquad (8)$$

where $\beta$ is the quantile decay ratio and $t$ is the current time. Intuitively, it becomes clear that all actors undergo initial optimization across the complete return distribution, after which they progressively fine-tune their optimization within their respective designated quantiles. Furthermore, it's worth noting that all actors share a common replay buffer. This buffer stores transitions $(s, a, r, s')$ stemming from interactions between all actors and the environment. These stored transitions are readily accessible for training the distributional critic.

## 3.3 FACILITATING DIVERSITY AND STABILITY FOR POPULATION

A population of actors learning across different quantiles of the return distribution simplifies the exploration process. This is because each actor focuses on a smaller subset of users, rendering the exploration task more effective. However, this approach brings forth new challenges. One challenge emerges when each actor within the population learns individually. This could lead to various members of the population sharing overlapping portions of the explored action space or continually switching between different behavioral patterns (Jackson & Daley, 2019; Parker-Holder et al., 2020). Such scenarios could potentially undermine the efficiency of the exploration process. Another challenge is the potential instability introduced by multiple actors learning simultaneously. Therefore, it becomes crucial to ensure that the actor population maintains both **Diversity** and **Stability**.

**Diversity.**    Following (Parker-Holder et al., 2020), we measure the diversity of the population using individual *behavior embeddings* to compute the volume of the kernel matrix between actors. By utilizing it as a regularization term, we prevent actors from becoming too similar, thereby promoting diversity and enabling a broader exploration of potential actions.

The behavior embedding, denoted as $\Phi(\pi) = \{\pi(\cdot|s)\}_{s \in \mathcal{S}}$, serves as a vectorized representation of an actor's behavior. This representation is embedded within the behavior space, facilitating the measurement of similarity between different actors. However, in the context of recommender systems, the state space is exceptionally vast and complex. To manage this challenge, a subset of states is sampled, with the number of samples significantly smaller than the size of the overall state space. Utilizing the subsequent formula, we approximate the behavior embedding through an expectation:

$$\widehat{\Phi}(\pi) = \mathbb{E}_{s \sim \mathcal{S}}\left[\{\pi(\cdot|s)\}\right]. \qquad (9)$$

In practice, we will sample a batch of states from the shared replay buffer of the population. We then execute all actors on these sampled states to approximate their respective behavioral embeddings. Once these approximations are obtained, we use a specific metric to quantify the diversity within the actor population. To compare two different actors, we employ a kernel function $\mathcal{K}$ to map

their behavioral embeddings into a higher-dimensional feature space. In this space, we compute a kernel matrix $\mathbf{K}$ among the actors within the population, defined as $\mathbf{K} = \mathcal{K}\left(\widehat{\Phi}\left(\pi_{\phi_i}\right), \widehat{\Phi}(\pi_{\phi_j})\right)_{i,j=1}^m$. Geometrically, the determinant of this matrix, $\det(\mathbf{K})$, corresponds to the volume of a parallelepiped defined by the feature maps of the chosen kernel function. A larger volume indicates a higher perceived degree of diversity within the actor population. We optimize population diversity as a regularization term in the population loss function. Specifically, this loss is defined as:

$$\mathcal{L}_{\text{div}}(\phi_1, \ldots, \phi_m) = -\log \det(\mathbf{K}) = -\log \det\left(\mathcal{K}\left(\widehat{\Phi}\left(\pi_{\phi_i}\right), \widehat{\Phi}(\pi_{\phi_j})\right)_{i,j=1}^m\right). \qquad (10)$$

**Stability.** To ensure stability in the learning process, we introduce a supervision module $\mathcal{L}_{\text{sta}}$. Its objective is to align the outputs of the actors with the user response. With the help of the supervision module, each actor can more effectively utilize the detailed user feedback on every item. One simple approach to achieve this is to combine the cross-entropy loss of each actor, defined as follows:

$$\mathcal{L}_{\text{sta}}(\phi_i) = \mathbb{E}_{(s,a)\sim\pi_{\phi_i}}\left[y \log \pi_{\phi_i}(a|s) + (1-y)\log\left(1 - \pi_{\phi_i}(a|s)\right)\right], \quad i \in \{1, \ldots, m\}, \qquad (11)$$

where $y$ is the supervision signal such as user clicks. Finally, the total population loss of UOEP can be written as:

$$\mathcal{L}_{\text{total}}(\phi_1, \ldots, \phi_m) = \sum_{i=1}^m \left(\mathcal{L}_{\text{actor}}(\phi_i) + \lambda_1 \mathcal{L}_{\text{sta}}(\phi_i)\right) + \lambda_2 \mathcal{L}_{\text{div}}(\phi_1, \ldots, \phi_m), \qquad (12)$$

where the coefficients $\lambda_1$ and $\lambda_2$ are coefficients dynamically tuned through a two-armed bandit framework. In this context, we consider the two losses, $\mathcal{L}_{\text{div}}$ and $\mathcal{L}_{\text{sta}}$, as the two arms of the bandit, and we assign reward probability distributions to decide which loss to optimize. These reward probability distributions are updated based on the observed recommendation performance at each time step, allowing us to strike a balance between emphasizing stability and diversity during the training process. For more in-depth implementation details, please refer to Appendix B.

## 4 EXPERIMENTS

In this section, we conduct experiments on public and industrial datasets to verify the effectiveness of the proposed UOEP method. We mainly focus on the following questions: *Q1. Can UOEP consistently outperform previous state-of-the-art methods (Section 4.2)? Q2. How does UOEP work and how do each of its components contribute (Section 4.3)? Q3. What potential does UOEP have (Section 4.4)?* The source code of experiments is shared at `https://anonymous.4open.science/r/UOEP-3A6A`.

### 4.1 EXPERIMENTAL SETTINGS

**Datasets.** To facilitate the testing of RL-based methods' long-term performance, we select three recommendation datasets: *KuaiRand1K*[1] is a recent dataset for sequential short-video recommendation, and we use the 1K version with irrelevant videos removed. *ML1M*[2] is a subset of the Movie-Lens dataset, which consists of 1 million user ratings of movies. *RL4RS*[3] is a session-based dataset that was introduced in the BigData Cup 2021 to promote recommendation research in RL. We pre-process them into the sequential recommendation format and split the data based on the recorded timestamps, with the first 75% used for training and the last 25% for evaluation. The preprocessing methods are described in Appendix C.1 and the statistics are provided in Table C.2.

**Online Environment Simulator.** Following (Liu et al., 2023), we construct online simulators based on the datasets to capture the reward signals obtained from interactions with users in each round. Specifically, we train a user response model $\Psi : \mathcal{S} \times \mathcal{A} \rightarrow \mathbb{R}^n$ for each dataset. The user state is derived from static user features and dynamic historical interactions. $\Psi$ outputs the probabilities

---

[1] https://kuairand.com/

[2] https://grouplens.org/datasets/movielens/1m/

[3] https://github.com/fuxiAIlab/RL4RS

that the user will provide positive feedback for each item in the recommended list $a_t$. The final user response, which is a binary vector $\mathbf{y}_t \in \{0, 1\}^n$ (e.g. click or no click), is then sampled uniformly from these probabilities.

**Evaluation Metrics and Baselines.** We evaluate long-term performance using two metrics: *Total Reward* (sum of rewards in a user session) and *Depth* (number of interactions in a session). These metrics are obtained by simulating user sessions in an online environment with the learned policy. Higher values indicate better performance in both metrics. Appendix C.3 provides detailed information on reward and session designs. We compare our method with several baselines, including supervised learning, classic reinforcement learning methods (A2C, DDPG, TD3), Wolpertinger method for large discrete action space, and exploration-based reinforcement learning method HAC. Appendix D and Appendix E contain detailed introductions and settings for all the algorithms.

## 4.2 OVERALL PERFORMANCE

Table 1: Overall performance of the proposed UOEP and all the baselines on three datasets. The best performance is shown in bold, second best performance is underlined. Experiments are repeated 5 times with different random seeds, and the average and standard deviation are reported.

| Algorithms | KuaiRand | | ML1M | | RL4RS | |
|---|---|---|---|---|---|---|
| | Total Reward | Depth | Total Reward | Depth | Total Reward | Depth |
| SL | 14.16±0.05 | 14.78±0.04 | 14.85±0.46 | 15.40±0.41 | 7.76±0.25 | 9.06±0.23 |
| A2C | 9.48±0.18 | 10.60±0.17 | 13.00±0.46 | 13.72±0.41 | 7.83±0.12 | 9.16±0.11 |
| DDPG | 10.66±2.98 | 11.66±2.66 | 15.34±0.68 | 15.83±0.61 | 7.77±0.95 | 9.07±0.88 |
| TD3 | 11.46±1.37 | 12.38±1.23 | 15.36±0.45 | 15.85±0.41 | 7.62±0.46 | 8.99±0.38 |
| Wolpertinger | 12.44±0.81 | 13.25±0.72 | 15.84±0.46 | 16.27±0.41 | 7.77±0.43 | 9.08±0.43 |
| HAC | 13.49±0.52 | 14.18±0.46 | 15.96±0.32 | 16.38±0.28 | 7.66±0.74 | 9.00±0.64 |
| UOEP | **14.39±0.10** | **14.98±0.09** | **16.59±0.11** | **16.95±0.10** | **8.48±0.53** | **9.60±0.50** |

We train UOEP with $m = 5$ actors and assign their quantile values $\alpha = 0.2, 0.4, 0.6, 0.8, 1.0$, respectively. For each model, we conduct a grid search on the hyperparameters to pick the setting with the best results and perform experiments with 5 random seeds, reporting the mean performance in Table 9. We can observe that our UOEP framework consistently achieves the best performance across all datasets. Compared to the best baselines, it improves performance by 1.6%, 3.9%, and 8.3% on three datasets, indicating the effectiveness of our proposed algorithm. Notably, on the KuaiRand dataset with the largest action space (11643 items), only our algorithm overperforms the supervised learning algorithm, demonstrating the stronger exploration capability in large action space.

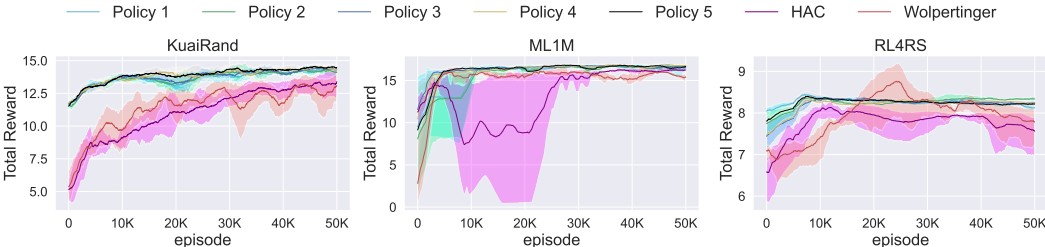

Figure 3: Learning curves for 5 actors of UOEP, HAC, and Wolpertinger on three datasets.

For RL-based baselines, A2C demonstrates excellent performance in RL4RS. However, it performs poorly on datasets with larger action space, such as KuaiRand and ML1M. On the other hand, DDPG improves the performance on these datasets by utilizing a deterministic policy. TD3 exhibits similar behavior to DDPG and slightly enhances the quality of recommendations. Wolpertinger, designed for large discrete action space, outperforms other classical reinforcement learning methods in KuaiRand and ML1M datasets. Moreover, compared with other baselines, HAC further enhances performance on datasets with large amount of items such as KuaiRand and ML1M, by emphasizing its exploration design. We also provide the learning curve of UOEP, HAC, and Wolpertinger in Figure 3. It can be observed that the learning of all five actors in UOEP is faster and more stable across all three datasets compared to the other two baseline methods. The learning processes of HAC and Wolpertinger exhibit significant fluctuations in ML1M and RL4RS, respectively, further demonstrating the effectiveness and stability of our user-oriented exploration approach.

### 4.3 FURTHER ANALYSIS AND ABLATION STUDIES

**How does UOEP work?** To understand the working process of UOEP, we plot t-SNE embeddings (Van der Maaten & Hinton, 2008) of the five actors learned by UOEP for generating actions. During the testing phase, we randomly sample a subset of users and serve them with the five learned actors. The generated actions are displayed in different colors. As shown in Figure 4, it can be observed that actions from each actor form distinct clusters. This clustering behavior serves as evidence that UOEP successfully explores a broad action space tailored for different users.

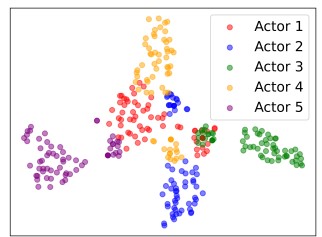

Figure 4: The t-SNE visualization of the population.

**Effect of the Distributional Critic.** In our method, we employ a distributional critic that allows actors to optimize at different quantile levels of cumulative rewards. To explore the effects of the distributional critic and the role of optimizing at different quantile levels of cumulative rewards, we disabled the distributional critic in UOEP and replaced it with a deterministic critic. In other words, all our actors optimized the expectation of the cumulative return distribution. We provide the recommendation quality of the distributional and deterministic critic in Table 2. We can observe a significant decrease in recommendation quality if we use the distributional critic, highlighting the importance of exploration in different user groups.

Table 2: Ablations for the distributional critic in UOEP on KuaiRand.

| Critic | Total Reward | Depth |
|---|---|---|
| Distributional | **14.39 ± 0.10** | **14.98 ± 0.09** |
| Deterministic | 13.01 ± 0.42 | 13.76 ± 0.37 |

**Number of Actors in UOEP.** The number of actors in UOEP has a significant impact on performance. To delve deeper into its effect, we conduct experiments to find the number that achieves the best performance. Specifically, we choose number of actors $m \in \{2, 3, 4, 5, 6\}$ and set the quantiles to $\{\frac{1}{m}, \frac{2}{m}, \dots, 1\}$ accordingly. The results are shown in Figure 5. It can be seen that with the

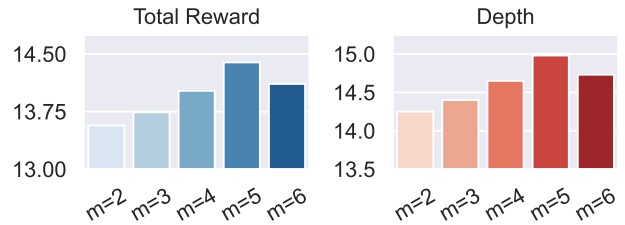

Figure 5: Ablations for the number of actors (denoted by $m$) in UOEP on KuaiRand.

increase in the number of actors, the quality of recommendation basically shows an upward trend, indicating that finer grouping of users ease the exploration process and improve the long-term performance. Notably, the performance achieves its peak when $m = 5$ and then declines, probably due to the increased difficulty of learning UOEP as the population size grows. Furthermore, since the training of each actor is independent, parallelizing UOEP is not difficult to implement, allowing us to easily accelerate the training speed.

**Effect of Regularization Loss in UOEP.** To explore the effects of the designed regularization losses, we disable $\mathcal{L}_{\text{div}}$, $\mathcal{L}_{\text{sta}}$ and both in UOEP, respectively. As shown in Table 3, removing any regularization loss leads to a significant decrease in performance, with removing both having an even larger negative impact. This indicates that improving both diversity and

Table 3: Ablations for the regularization losses in UOEP on KuaiRand.

| | Total Reward | Depth |
|---|---|---|
| UOEP | **14.39 ± 0.10** | **14.98 ± 0.09** |
| w/o $\mathcal{L}_{\text{div}}$ | 14.03 ± 0.20 | 14.66 ± 0.18 |
| w/o $\mathcal{L}_{\text{sta}}$ | 13.11 ± 0.62 | 13.85 ± 0.56 |
| w/o $\mathcal{L}_{\text{div}}, \mathcal{L}_{\text{sta}}$ | 11.50 ± 0.90 | 12.41 ± 0.81 |

stability in exploration is beneficial for the performance of our algorithm, resulting in higher recommendation quality. Furthermore, removing the stability loss resulted in a marked increase in variance, highlighting its role in minimizing instability during the learning process.

### 4.4 EXPLORING UOEP'S POTENTIAL: LOW-ACTIVITY USER EXPERIENCE AND FAIRNESS

Our actor training leverages the CVaR values from the distributional critic, which focuses on the tail-end of the return distribution. Inspired by recent work that integrates CVaR with collaborative filtering to enhance the experiences of low-activity users (Togashi et al., 2023), we hypothesize

Table 4: Low-Activity users and fairness performance.

| Algorithms | KuaiRand | | | ML1M | | | RL4RS | | |
|---|---|---|---|---|---|---|---|---|---|
| | $CVaR_{0.3}$ | $CVaR_{0.4}$ | Gini(%) | $CVaR_{0.3}$ | $CVaR_{0.4}$ | Gini(%) | $CVaR_{0.3}$ | $CVaR_{0.4}$ | Gini(%) |
| SL | 1.72 | 5.40 | 26.81 | 5.64 | 8.08 | 21.28 | 2.56 | 4.62 | 19.04 |
| A2C | -0.15 | 0.08 | 45.22 | 4.61 | 6.50 | 24.24 | 1.95 | 3.61 | 24.17 |
| DDPG | 0.06 | 1.43 | 36.56 | 6.57 | 8.98 | 19.69 | 2.44 | 4.20 | 20.61 |
| TD3 | 0.19 | 1.57 | 37.56 | 6.87 | 9.24 | 19.09 | 2.87 | 4.20 | 21.27 |
| Wolpertinger | 0.19 | 2.05 | 34.32 | 7.67 | 10.00 | 17.66 | 2.69 | 4.25 | 20.88 |
| HAC | 0.26 | 2.30 | 33.82 | 7.89 | 10.23 | 17.25 | 1.82 | 3.39 | 25.16 |
| UOEP (OURS) | **2.48** | **6.01** | **25.67** | **9.51** | **11.59** | **14.64** | **5.94** | **6.76** | **10.54** |

that UOEP has similar capabilities for long-term performance, which we validate through empirical analysis. Furthermore, we explore UOEP's potential to improve the fairness of the long-term performance within the recommender system. We conduct experiments across 3 datasets with 5 random seeds. To analyze the model's performance on low active users, we choose $CVaR_{0.3}$ and $CVaR_{0.4}$ as the metrics, which represent the average value of the lower 30% and 40% of the Total Reward distribution over the test set, respectively. For assessing fairness, we utilize the Gini coefficient (Wang et al., 2023) as our metric, with lower values indicating greater fairness. Please refer to Appendix F for more details of Gini coefficient. As summarized in Table 4, UOEP outperforms on both the $CVaR_{0.3}$ and $CVaR_{0.4}$ metrics across all three datasets. Additionally, the model achieves significant reductions in the Gini coefficient. These results solidly establish UOEP's effectiveness in serving low-activity users and promoting long-term fairness within the system.

## 5 RELATED WORK

**Reinforcement Learning in Recommender Systems.** Deep reinforcement learning, combining deep neural networks with reinforcement learning, has gained attention in recommender systems research. Early works like (Shani et al., 2005) formulated recommendation as an MDP and experimented with model-based RL. Zheng et al. (2018) first applied DQN for news recommendation. Dulac-Arnold et al. (2015) enabled RL for large discrete action space. Liu et al. (2018) tested actor-critic methods on recommendation datasets. Recently, RL has shown success in real-world applications. Chen et al. (2019) scaled batch RL to billions of users. Hu et al. (2018) extended DDPG for learning-to-rank. Liu et al. (2023) proposed aligned hyper actor-critic learning in large action space. Chen et al. (2023) enabled adaptive re-ranking with multi-objective RL without retraining. Overall, deep RL has become an important technique for building recommender systems.

**Exploration in Reinforcement Learning.** Balancing exploitation and exploration is a key challenge in reinforcement learning. Classic strategies include epsilon-greedy exploration (Sutton & Barto, 1999), parameter space noise (Plappert et al., 2017), upper confidence bound algorithms (Auer et al., 2002), intrinsic motivation techniques like count-based bonuses (Bellemare et al., 2016) and prediction error (Burda et al., 2018). Novelty search rewards reaching new states regardless of external reward (Lehman & Stanley, 2011). Recent methods optimize exploration by adapting behavioral diversity (Parker-Holder et al., 2020) or information-theoretic bonuses (Houthooft et al., 2016; Mohamed & Jimenez Rezende, 2015; Shyam et al., 2019). Overall, many techniques have been developed to address the exploration challenge in different ways.

## 6 CONCLUSION

In this work, we propose UOEP to reinforce user-oriented exploration in recommender systems. UOEP works by first characterizing the activity level of users based on the return distribution under different quantiles. It then learns multiple actors where each actor corresponds to a specific user group with a predefined level of activity. Thus, UOEP can customize the exploration intensity for different user groups. Moreover, a population diversity regularization along with a supervision term is designed to ensure diversity and stability during the actor learning process. We conduct extensive experiments on various public and industrial recommendation datasets. Experimental results demonstrate the advantages of UOEP over previous state-of-the-art algorithms in long-term performance. Further analyses indicate enhanced experience for low-activity users and improved fairness.

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

# A  OVERALL ALGORITHM

We train an online environment simulator and choose the DDPG algorithm as the backbone of our UOEP algorithm.

**Training Phase.**   We provide the training phase of UOEP in Algorithm 1. Particularly, we assign a target network to both the critic and each actor. In each epoch, we calculate the critic's loss (line $10-12$), each actor's loss (line $14-17$), population diversity loss ($18-20$) and supervised loss (line $21$), respectively, and perform gradient descent and soft update (line $13, 23-24$).

**Inference Phase.**   Inference phase of UOEP is show in Algorithm 2. Only the trained distributional critic and actor population are used during execution. Each time the recommender system receives the user's status, we use a *critic-trusted* method to select the optimal action. Specifically, all actors need to provide action $a$ to the critic, and then the critic selects the one with the largest return expectation to execute. The calculation of return expectation is shown in Eq. 13.

$$Q(s, a) = \mathbb{E}\left[\mathcal{Z}(s, a)\right] \approx \frac{1}{K} \sum_{k=1}^{K} Z\left(s, a; \tau_k\right), \tau_k \sim \mathbb{U}(0, 1). \tag{13}$$

---

**Algorithm 1** Training phase of UOEP

---

**Input:**  Shared replay buffer $\mathcal{B} = \{(s, a, r, s')\}$; Critic $Z_\theta$ and critic-target $Z_{\theta'}$; Population $\mathcal{P}$ with size $m$ includes actors $\pi_{\phi_1}, \ldots, \pi_{\phi_m}$, actor-targets $\pi_{\phi_1'}, \ldots, \pi_{\phi_m'}$ and corresponding quantile levels $\alpha_1, \ldots, \alpha_m$; environment $\mathcal{E}$; Online simulator $\mathcal{S}$; Critic-loss parameters $N, N', \kappa$, learning rate $\eta$, soft update parameter $\mu$, quantile decay ratio $\beta$, diversity trade-off parameter $\lambda_1$, supervised trade-off parameter $\lambda_2$.

**Output:**  Population $\mathcal{P}^*$ includes $m$ actors $\pi_{\phi_1}^* \ldots \pi_{\phi_m}^*$ learned from their each quantile level $\alpha_i$.

1: Randomly Initialize critic network $Z_\theta$ and actors $\pi_{\phi_1}, \ldots, \pi_{\phi_m}$.
2: Initialize target network $Z_{\theta'} \leftarrow Z_\theta, \pi_{\phi_1}', \ldots, \pi_{\phi_m}' \leftarrow \pi_{\phi_1}, \ldots, \pi_{\phi_m}$.
3: Initialize replay buffer $\mathcal{B}$ and environments $\mathcal{E}$.
4: **for** $t = 1, \ldots$ **do**
5:   **for** $i = 1, \ldots, m$ **do**
6:     Select action $a_{t,i} = \pi_{\phi_i}(s_{t,i})$ according to the current actor and observation state $s_{t,i}$.
7:     Execute action $a_i^t$ and get reward $r^t$ from simulator $\mathcal{S}$ then observe new state $s_i^{t+1}$.
8:     Store transition $(s_{t,i}, a_{t,i}, r_{t,i}, s_{t+1,i})$ in $\mathcal{B}$.
9:   **end for**
10:   Sample a random minibatch of $B$ transitions $(s_i, a_i, r_i, s_{i+1})$ from $\mathcal{B}$.
11:   Sample $N$ quantiles $\tau$ and $N'$ target quantiles $\tau'$ from $\mathcal{U}(0, 1)$ and compute $\delta_{\tau, \tau'}$ in Eq.3.
12:   Compute critic loss $\mathcal{L}_{\text{critic}}(\theta)$ by Eq.5.
13:   Gradient step $\theta \leftarrow \theta - \eta \mathcal{L}_{\text{critic}}(\theta)$.
14:   **for** $i = 1, \ldots, m$ **do**
15:     Compute decayed quantile level $\alpha_{t,i}$ in Eq.8 with $\alpha_i$ and $t$.
16:     Compute actor loss $\mathcal{L}_{\text{actor}_i}(\phi_i)$ in Eq.7.
17:   **end for**
18:   Sample a random minibatch of states $s$ from $\mathcal{B}$.
19:   Compute behavioral embeddings $\widehat{\Phi}\left(\pi_{\phi_1}\right), \ldots, \widehat{\Phi}\left(\pi_{\phi_m}\right)$ in Eq.9 with $s$.
20:   Compute population diversity loss $\mathcal{L}_{\text{div}}(\phi_1, \ldots, \phi_m)$ in Eq.10.
21:   Compute supervised loss $\mathcal{L}_{\text{sta}}(\phi_1, \ldots, \phi_m)$ in Eq.11.
22:   Compute total population loss $\mathcal{L}_{\text{total}}(\phi_1, \ldots, \phi_m)$ in Eq.12.
23:   Gradient step $\phi_1, \ldots, \phi_m \leftarrow \phi_1, \ldots, \phi_m - \eta \nabla \mathcal{L}_{\text{total}}(\phi_1, \ldots, \phi_m)$.
24:   Perform soft-update on $\theta' \leftarrow \mu\theta + (1 - \mu)\theta'$; $\phi' \leftarrow \mu\phi + (1 - \mu)\phi'$.
25:   Update $\lambda_1$ and $\lambda_2$ with two-armed bandit with cumulative rewards get from $\mathcal{E}$.
26: **end for**
27: **return** $\mathcal{P}^*$.

---

---

**Algorithm 2** Inference phase of UOEP

---

**Input:** Critic $Z_\theta$; Population $\mathcal{P}$ includes actors $\pi_{\phi_1}, \ldots, \pi_{\phi_m}$; Current state $s$.
**Output:** Optimal action $a^*$
 1: **for** $t = 1, \ldots, m$ **do**
 2:     Select action $a_i = \pi_{\phi_i}(s)$ according to the current actor and observation state $s$.
 3:     Calculate $Q(s, a_i)$ through Eq.13.
 4: **end for**
 5: Choose optimal action according to $a^* = \arg\max_{a_i} Q(s, a_i), i \in \{i, \ldots, m\}$.
 6: **return** $a^*$.

---

## B   BALANCING STABILITY AND DIVERSITY IN UOEP

We use a two-armed bandit approach to adaptively select the loss function to be optimized, in order to encourage either stability or diversity during different stages of optimization. Specifically, during each training iteration, we choose between the diversity loss and the stability loss for the model. These two loss functions represent different training objectives. If we observe signs of deteriorating performance, indicating a decline in model performance, we select the stability loss to reduce the impact of excessive exploration and make the model more stable. Conversely, if we observe signs of improving performance, indicating an enhancement in model performance, we select the diversity loss to increase the model's exploration behavior and search more extensively for potential optimization directions.

We employ a bandit algorithm based on Thompson sampling, where the optimal policy is to pull the arm with the highest average reward (Agrawal & Goyal, 2012). Specifically, the Thompson sampling algorithm initially assigns a beta prior distribution to each arm. Then, prior to each arm selection, an independent sample is drawn from the beta distribution of each arm, representing its posterior probability distribution, and the arm with the highest sample value is chosen as the currently selected arm. Specifically, by setting a hyperparameter $\lambda$ as the regularization loss coefficient, if we choose the first arm, we assign $\lambda_1$ as $\lambda$ and $\lambda_2$ as 0. Conversely, if we choose the second arm, we assign $\lambda_1$ as 0 and $\lambda_2$ as $\lambda$. After pulling the selected arm, we receive a reward defined as an indicator variable, $r_t = \mathbb{1}(R_{t+1} > R_t)$, where $R_t$ represents the reward observed at time $t$, and $R_{t+1}$ represents the reward observed at time $t + 1$. We then use this reward to update the parameters of the beta distribution for each arm. If a positive reward is obtained, the corresponding arm's prior parameters are increased. If a negative reward is obtained, the corresponding arm's prior parameters remain unchanged. Through iterative sampling, selection, and parameter updating, the bandit can balance diversity and stability during the training process of UOEP, ultimately finding the optimal action policy.

## C   ENVIRONMENT SETUP

### C.1   DATASET PREPROCESSING

The three datasets are preprocessed into a unified format, arranging each record in chronological order to include user features, user history, exposed items, user feedback, and timestamps. Then we split them into the first 75% for training and the last 25% for evaluation according to record timestamps.

Similar to the dataset preprocessing in (Liu et al., 2023), for the ML1M dataset, we consider movies rated 4 and 5 by users as positive samples, and other movies as negative samples. For the KuaiRand dataset, we first remove videos with less than 50 occurrences and then consider videos with a watch time ratio greater than 0.8 as positive samples, and others as negative samples.

For both the ML1M and KuaiRand datasets, we split each user session into sequences of item lists with a length of 10, in chronological order. Only the positive samples before each segmented list are considered historical behavior. The formatted data follows the same structure as the RL4RS dataset.

## C.2 STATISTICS OF DATASETS AFTER PREPROCESS

The statistics of datasets after preprocessing are presented in Table 5. However, it should be noted that the RL4RS dataset provides user profile features instead of user IDs, so it does not have a count of unique users in the dataset.

Table 5: Statistics of datasets after preprocessing.

| Dataset | Users | Items | # of record | List size $n$ |
|---|---|---|---|---|
| KuaiRand | 986 | 11,643 | 96,532 | 10 |
| MovieLens-1M | 6041 | 3953 | 97,382 | 10 |
| RL4RS | - | 283 | 781,367 | 9 |

## C.3 REWARD AND SESSION DESIGNS

In each round of the Session, the recommender system provides the user with a list of items, and the user provides feedback. Our reward function, $r(s_t, a_t)$, is designed as the average reward obtained from all items in the list. Specifically, items that are clicked receive a reward of 1, while items that are not clicked receive a reward of $-0.2$.

Additionally, each user is assigned an initial temper value at the beginning of the session. In each interaction, the temper value is reduced to varying degrees based on the quality of the recommendation. If the recommendation quality is poor, the user's temperament value decreases rapidly until it reaches 0 or lower, which is considered the end of the user session. Furthermore, there is a maximum depth limit of 20 rounds for each session.

## D BASELINES

Our method is compared with various baselines, including supervised learning method, classic reinforcement learning methods (A2C, DDPG, TD3), and Wolpertinger, HAC:

- **Supervise Learning:** The model is optimized using the observed exposure (recommended item lists) and corresponding user feedback from the training set through a binary cross-entropy loss function on the offline training set.
- **A2C:** A2C (Mnih et al., 2016) is a reinforcement learning algorithm that combines the advantages of both policy gradients and value-based methods by using an actor-critic architecture to learn policies and value functions simultaneously.
- **DDPG:** DDPG (Lillicrap et al., 2015) is an off-policy actor-critic algorithm that can handle continuous action space, using a deterministic policy gradient approach to learn a deterministic policy and value function.
- **TD3:** TD3 (Fujimoto et al., 2018) is an enhancement of the DDPG algorithm that introduces twin critics and delayed updates to improve stability and performance in continuous control tasks, particularly in handling overestimation bias of the value function.
- **Wolpertinger:** Wolpertinger (Dulac-Arnold et al., 2015) is a method that embeds a large number of discrete actions into a continuous space and utilizes approximate nearest-neighbor methods for action selection, enabling efficient reinforcement learning in environments with a large number of discrete actions.
- **HAC:** HAC (Liu et al., 2023) introduces a hyper-actor and critic learning framework that decomposes the item list generation process into two steps: hyper-action inference and effect-action selection, incorporating an alignment module and a kernel mapping function for ensuring alignment between the action space.

## E EXPERIMENTAL DETAILS

### E.1 ARCHITECTURES

**Actor.** All RL-based methods use SASRec (Kang & McAuley, 2018) as the backbone for the actor. Given the user's historical items, the item encoder is used to map them to a 32-dimensional vector

and adds trainable positional embeddings. This is then inputted into a 2-layer Transformer encoder with 4 heads and a dropout rate of 0.1, resulting in an output action vector. Then, the action vector is dot-producted with the embeddings of all items, and the top-k items are selected to be provided to the user.

**Deterministic Critic.** As the critic for Deterministic methods like DDPG and TD3, it is an MLP that takes the encoded user state vector and action vector as inputs. It consists of two hidden layers with dimensions of 256 and 64, and outputs a scalar representing the Q-value.

**Value Critic.** As the critic for A2C, it is an MLP that takes the encoded user state vector as input. It also has two hidden layers with dimensions of 256 and 64, and outputs a scalar representing the Q-value. The difference is that in A2C, the value network takes only the user state as input.

**Implicit Quantile Network.** As the critic for UOEP, it takes the encoded user state vector, action vector, and quantile values as inputs. The user state vector and action vector pass through two hidden layers with dimensions of 256 and 64, resulting in a 16-dimensional vector. This vector is then multiplied element-wise with a 16-dimensional vector obtained from a single MLP layer applied to the quantiles. The resulting vector is passed through a hidden layer with a dimension of 32 to output the quantile values.

### E.2 HYPERPARAMETERS

In all methods and experiments, we conducted a grid search on the hyperparameters to pick the setting with the best results. The hyperparameters for the UOEP setting on the KuaiRand dataset are provided in Table 6. All methods were implemented using PyTorch.

Table 6: Hyper-parameters of UOEP.

| Hyper-parameter | Value |
| --- | --- |
| Optimizer | Adam (Kingma & Ba, 2014) |
| Actor Learning Rate | $5 \times 10^{-4}$ |
| Critic Learning Rate | $1 \times 10^{-3}$ |
| Target Update Rate | $1 \times 10^{-2}$ |
| Batch Size | 64 |
| Gradient Clipping | False |
| Number of Epoch | 50000 |
| $\gamma$: Discount Factor | 0.9 |
| $M$: Population Size | 5 |
| $\alpha_1, \ldots, \alpha_m$: Quantile Levels for Each Policy | $0.2, 0.4, 0.6, 0.8, 1.0$ |
| $N$: Sample Number of Target quantile for Critic Loss | 32 |
| $K$: Sample Number of Target quantile for Actor Loss | 8 |
| $\beta$: quantile Decay Ratio | 0.5 |
| $\lambda$: regularization loss coefficient | 16 |

## F   FAIRNESS METRIC

**Gini coefficient.** The Gini coefficient is a widely used measure in sociology and economics to assess social inequality (Fu et al., 2020; Leonhardt et al., 2018; Ge et al., 2021; Mansoury et al., 2020). It is also commonly employed as a metric for individual fairness in recommender systems. The Gini coefficient can be applied to evaluate fairness based on various factors, such as predicted user relevance (Fu et al., 2020; Leonhardt et al., 2018) or item exposure (Ge et al., 2021; Mansoury et al., 2020). A lower Gini coefficient indicates a fairer distribution of recommendations, implying a more equitable allocation of relevant items or exposure across users or items. The formal formulation of the Gini coefficient is as follows:

$$\text{Gini} = \frac{\sum_{v_x, v_y \in \mathcal{V}} \left| f(v_x) - f(v_y) \right|}{2|\mathcal{V}| \sum_v f(v)}. \tag{14}$$

In our setting, $\mathcal{V}$ represents all the data in the test set, and $f$ represents the mapping from each data point to its *Total Reward*.

# G  FURTHER ANALYSIS

## G.1  REEVALUATING USER ENGAGEMENT: CTR AND CLICK NUMBER

While a high click-through rate (CTR) is often interpreted as a marker of user engagement, it may not always provide an accurate representation of user activity levels. Such discrepancies could undermine the reliability of our validation experiments. To address this concern, we conducted a detailed analysis of the relationship between CTR and total click counts within our KuaiRand dataset, as depicted in Figure 6. This investigation revealed a notable trend: users with fewer total clicks tend to have lower CTRs on average.

Recognizing the potential limitations of relying solely on CTR, we revisited our validation experiments, this time categorizing participants based on their total number of clicks. The reanalyzed results, illustrated in Figure 7, demonstrate a clear correlation between user activity levels and their preferences for different types of content, aligning with our prior observations. This supplementary analysis not only reinforces our initial conclusions but also validates that our findings are consistent when examined through alternate metrics.

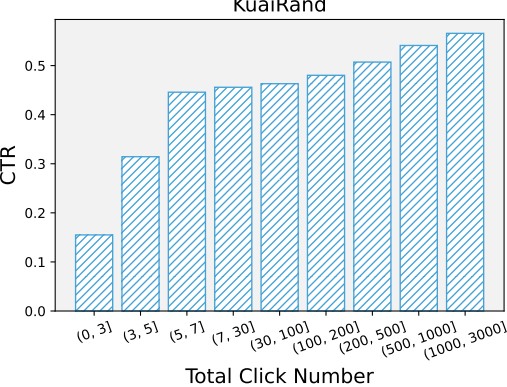

Figure 6: Relationship between CTR and the total number of clicks on KuaiRand.

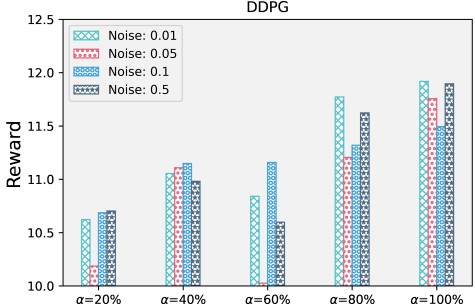
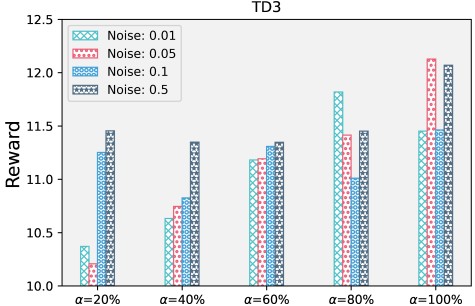

Figure 7: Illustration Experiment For Reevaluation. We resorted users based on their activity levels (total number of clicks) and selected the five bottom $\alpha$-quantile ($\alpha \in \{0.2, 0.4, 0.6, 0.8, 1.0\}$) of user groups. We then trained two RL algorithms, DDPG and TD3, on these five groups under four varying noise levels. We conducted all experiments with four different seeds and reported and averaged the results.

## G.2 ASSESSING UOEP'S POTENTIAL: ACROSS DIFFERENT USER ACTIVITY LEVELS

In Section 4.4, our discussion was primarily centered around the enhanced performance of the UOEP method for users with lower levels of activity. However, it's crucial to also consider its effectiveness among highly active users. To this end, we have extended our experimental scope to include high-activity users, incorporating these new findings in Table 7. Here, we showcase the average values of the top 40% and 50% of the Total Reward distribution, denoted as $ATR_{0.4}$ and $ATR_{0.5}$, evaluated across our test set.

It is pertinent to mention that the maximum possible return in our experimental setup is limited to 20, reflecting the cap we set on user-item interactions per session. Upon reviewing these updated results, it is evident that our method surpasses most baselines across three distinct datasets. An exception is noted in the RL4RS dataset, where it ranks second-best.

Nonetheless, it's important to recognize that the level of improvement observed for high-activity users, as depicted in these additional experiments, might not be as pronounced as those detailed in Section 4.4. This variation can be attributed to the fundamental design of our method, which prioritizes exploration — a feature that inherently benefits users with lower activity more substantially. Therefore, while our approach demonstrates efficacy across the entire spectrum of user activity, its most significant impact is observed amongst those with comparatively lower engagement levels.

Table 7: High-Activity users performance. Experiments are repeated 5 times with different random seeds, and the average is reported.

| Algorithms | KuaiRand | | ML1M | | RL4RS | |
|---|---|---|---|---|---|---|
| | $ATR_{0.4}$ | $ATR_{0.5}$ | $ATR_{0.4}$ | $ATR_{0.5}$ | $ATR_{0.4}$ | $ATR_{0.5}$ |
| SL | 20.00 | 20.00 | 19.91 | 19.75 | 10.35 | 10.13 |
| A2C | 18.94 | 17.83 | 18.46 | 17.94 | 11.06 | 10.85 |
| DDPG | 19.14 | 18.29 | 19.99 | 19.90 | 10.14 | 10.05 |
| TD3 | 19.70 | 19.16 | 19.86 | 19.77 | 9.66 | 9.44 |
| Wolpertinger | 20.00 | 19.96 | 20.00 | 19.99 | 10.50 | 10.29 |
| HAC | 20.00 | 19.89 | 20.00 | 19.99 | 9.91 | 9.80 |
| UOEP | 20.00 | 20.00 | 20.00 | 20.00 | 10.85 | 10.78 |

## G.3 UOEP GROUPING STRATEGY: BALANCING USER ACTIVITY LEVELS AND Q-VALUE ACCURACY

UOEP's strategy of dividing users into groups based on bottom $\alpha$ values, where $\alpha \in [0.2, 0.4, 0.6, 0.8, 1.0]$, results in a nested structure among these groups. For instance, the group with $\alpha = 0.4$ encompasses users from the $\alpha = 0.2$ group. This intentional overlap in grouping is underpinned by two fundamental reasons:

**Focus on Low-Activity Users.** As discussed in the introduction, we can directly exploit the well-established preferences of high-activity users more effectively due to the richer data available. Conversely, low-activity users stand to gain more from high-intensity exploration, as it helps uncover their latent interests. This observation is validated by our validation experiments in the introduction section. In our method, we specifically target the lower activity users by grouping them based on the bottom quantiles of the return distribution. This approach ensures that our system consistently prioritizes the exploration of interests among these users across different actors in the model. As a result, no matter which actor is served during the inference phase, the interests of low-activity users are always taken into account. By targeting the bottom quantiles of the return distribution, we ensure a concentrated focus on low-activity users, thereby capturing their interests more effectively.

**Overestimation of Q-Values.** Q-value overestimation in Reinforcement Kuznetsov et al. (2020); Duan et al. (2021); Prabhakar et al. (2022); Van Hasselt et al. (2016); Li & Hou (2019) Learning refers to the tendency of Q-learning algorithms to overestimate the value of actions, particularly in environments with noise and uncertainty. This can lead to suboptimal policies and erratic learning behavior. In our case, when users are grouped into multiple non-overlapping groups, this issue intensifies due to the challenge of accurately evaluating Q-values in more narrowly defined user groups. Such overestimation leads to incorrect assessments of the Q-value, which can impede convergence, destabilize learning, and reduce the efficiency of exploration.

To empirically validate our method, we compared the performance of our nested grouping model with a version using non-overlapping groups. The findings, detailed in Table 8, reveal a marginal performance edge in favor of the nested model. Furthermore, Figure 8 illustrates the Q-value distributions for both models, clearly showing the heightened overestimation in the non-overlapping version.

In conclusion, while a non-overlapping group approach might seem more straightforward and less redundant, our nested grouping strategy is deliberately chosen. It not only places a targeted emphasis on low-activity users but also effectively counters the challenges of Q-value overestimation. This approach ultimately leads to more effective and stable exploration policies.

Table 8: Overall performance of a non-overlapping version of UOEP against our original (nested) version of UOEP. Experiments are repeated 5 times with different random seeds, and the average and standard deviation are reported.

| Algorithms | KuaiRand | | ML1M | | RL4RS | |
|---|---|---|---|---|---|---|
| | Total Reward | Depth | Total Reward | Depth | Total Reward | Depth |
| UOEP (w/o overlap) | 14.37±0.26 | 14.97±0.24 | 16.60±0.06 | 16.96±0.52 | 8.24±0.31 | 9.43±0.28 |
| UOEP (ours) | 14.39±0.10 | 14.98±0.09 | 16.59±0.11 | 16.95±0.10 | 8.48±0.53 | 9.60±0.50 |

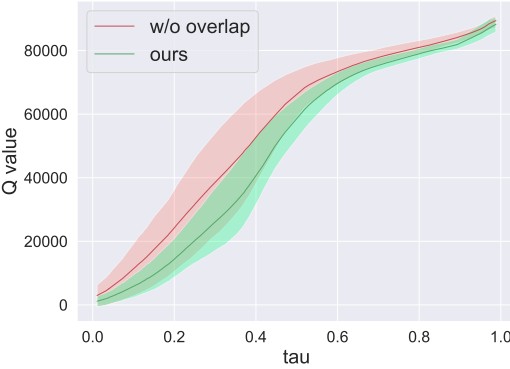

Figure 8: Q-value estimation. We randomly sample an initial state, take all items (11643 in total) as actions, and then sample 100 tau values between 0 and 1 respectively into the distributional critic of the trained UOEP (w/o overlap) and UOEP (original), and then take the average over all actions. We have tested for all five random seeds. The horizontal axis is tau, and the vertical axis is the q value estimated by the critic.

## H    FURTHER DISCUSSION

### H.1    USER-ORIENTED EXPLORATION IN RECOMMENDER SYSTEMS

In this section, we delve deeper into the unique contribution of our work within the realm of user-oriented exploration in recommender system research. The design of exploration strategies for users with varying activity levels is a well-trodden area, particularly highlighted in cold-start Zhang et al. (2023); Briand et al. (2021) and active learning Bu & Small (2018); Karimi et al. (2011) domains. These domains have extensively documented strategies catering to new users Rashid et al. (2002), those experiencing the cold-start phenomenon Zhang et al. (2023), or users with low engagement Carraro (2020), focusing on criteria such as activity level, popularity, and prediction uncertainty Merialdo (2001); Bu & Small (2018).

However, in this paper the use of the term "exploration" is specifically framed within reinforcement learning-based recommender systems (RL-based RS). It significantly differs from the methods employed in cold-start and active learning domains. In RL-based RS, the primary focus is on enhancing users' long-term engagement. This is achieved through an interactive learning paradigm that dynamically adjusts to user feedback. Unlike traditional recommender systems that may prioritize immediate user preferences, RL-based RS are designed to discover and cater to evolving user interests over time. This involves a balance between exploration and exploitation, where the system

not only leverages existing user data to provide relevant recommendations (exploitation) but also ventures into less-charted territories to uncover potential interests (exploration).

A notable gap in current methods for active learning and cold-start problems is their limited applicability within this interactive framework. The innovation of our work lies in customizing exploration at the user level within an RL context. This approach marks a significant departure from the traditional application of user exploration strategies in cold-start scenarios and active learning research. Moreover, existing RL-based RS methods Liu et al. (2023); Zhu & Van Roy (2023); Yan et al. (2023); Chen et al. (2021; 2019) have largely overlooked user-specific exploration, often employing data-independent exploration strategies or using cold-start metrics as simple reward signals. In contrast, our approach proposes user-oriented exploration strategies, a dimension less explored in the current RL-based RS literature.

## H.2 Between Exploration and Long-term User Experience

In this paper, we focus on exploration as a key factor in enhancing long-term user experiences, aligning with the prevalent approaches in RL and RL-based recommender systems. Exploration is commonly utilized in these fields to maximize long-term rewards.

For instance, diversity-based RL methods Hartikainen et al. (2019); Cideron et al.; Eysenbach et al. (2018); Parker-Holder et al. (2020) use exploration to improve the final returns. Likewise, RL-based recommender systems that emphasize exploration Liu et al. (2023); Chen et al. (2021); Zhu & Van Roy (2023); Yan et al. (2023) aim to increase long-term user engagement. Notably, large-scale experiments by Google Chen et al. (2021) have shown that exploration can significantly enhance user retention and activity levels. These results underscore the importance of integrating RL's exploration capabilities in recommender systems to improve long-term user experiences. Consequently, our baseline selection predominantly includes traditional and exploration-focused RL methods. We evaluate these methods using metrics of total reward and depth, which are indicative of long-term returns. This distinguishes our study from others that mainly address diversity or cold-start problems in recommender systems.

Despite the limitations in testing our method in real-world online environments, our offline simulation experiments offer a comprehensive comparison of various RL-based methods. These experiments demonstrate the efficacy of our approach, particularly in enhancing user engagement over extended periods.

Additionally, while our paper primarily does not delve into issues like diversity or the cold-start problem, we acknowledge that there may exist some relation between them and exploration in the realm of recommender systems. To explore our model's capabilities in these areas, we have included supplementary experimental results in Table 9 for evaluating coverage and diversity. For the diversity metric, we chose Intra-List Similarity (ILS) (Ziegler et al., 2005) as the diversity metric. The specific definition is as follows:

$$\text{ILS}(R) = \frac{2}{k(k-1)} \sum_{i \in R} \sum_{j \neq i \in R} \text{Sim}(i, j).$$

Where $R$ is the list, $k$ is the size of the list, and for the similarity function Sim, we chose a simple indicator function, that is, if two items are in the same category, it is 1, otherwise it is 0. Note that the smaller this metric, the better. These results highlight the superior coverage and diversity achieved by our method compared to two other exploration-oriented baselines, with the exception of a second-best performance of ILS in the RL4RS dataset.

Table 9: Coverage and Diversity of UOEP against other exploration-based baselines. Experiments are repeated 5 times with different random seeds, and the average is reported.

| Algorithms | KuaiRand | | ML1M | | RL4RS | |
|---|---|---|---|---|---|---|
| | Coverage | ILS | Coverage | ILS | Coverage | ILS |
| Wolpertinger | 0.00076 | 0.60 | 0.00078 | 0.46 | 0.00078 | 0.47 |
| HAC | 0.00082 | 0.65 | 0.00081 | 0.31 | 0.00082 | 0.59 |
| UOEP | 0.00122 | 0.48 | 0.00102 | 0.24 | 0.00108 | 0.52 |

