# OpenReview forum: "UOEP: User-Oriented Exploration Policy for Enhancing Long-Term User Experiences in Recommender Systems"
_ICLR.cc/2024/Conference — Submitted to ICLR 2024_

### Official Review · Reviewer_3uff · 2023-10-30

**Soundness:** 2 fair
**Presentation:** 2 fair
**Contribution:** 2 fair
**Rating:** 3
**Confidence:** 4

**Summary:**

This paper studies the user exploration problem in the recommender system. Different from previous works that adopt a uniform exploration policy for all users, this paper proposes a user-oriented exploration policy to learn different exploration strategies for different types of users. Specifically, this paper applies the risk-averse distributional reinforcement learning to maximize $CVaR_{\alpha}$. Moreover, the authors divide users into different groups based on the quantile score of expected returns and utilize population-based reinforcement learning to learn separate agents to optimize $CVaR_{\alpha}$ with different quantile scores $\alpha$. Experiments are conducted on the recommender simulator based on three real-world datasets.

**Strengths:**

1.	User exploration in the recommender system is an important problem.
2.	The structure of this paper is well-organized and easy to follow.
3.	The authors evaluate the proposed method using a recommender simulator based on three real-world datasets, which is comprehensive.

**Weaknesses:**

1.	To design a user-oriented exploration policy for different types of users, the authors divide users into different groups by the $CVaR_{\alpha}$ with $\alpha \in [0.2, 0.4, 0.6, 0.8, 1.0]$. According to Eq. (1), there is a nested relation between these five user groups. For example, the user group with $\alpha = 0.4$ contains the users in the user group with $\alpha = 0.2$. This definition is problematic and will result in a redundancy in policy optimization for different user groups.
2.	The motivation of this paper is to design a separate user-oriented exploration policy for different user groups. However, to my understanding, there is no explicit exploration strategy design for different user groups, and only the optimization objective $CVaR_{\alpha}$ varies for different user groups, which does not necessarily promote user exploration for different groups.
3.	The used evaluation metrics (total reward and Depth) do not validate the effectiveness of exploration. Other exploration-related evaluation metrics such as diversity and coverage are necessary to demonstrate the exploration performance.

**Questions:**

See the Weaknesses for the questions.

---

> ### Author Response · Authors · 2023-11-15
> **Response to Reviewer 3uff, Part 1**
>
> We appreciate your detailed and valuable feedback on our paper.
> > W1. To design a user-oriented exploration policy for different types of users, the authors divide users into different groups by the $\operatorname{CVaR}_\alpha$ with $\alpha \in [0.2,0.4,0.6,0.8,1.0]$. According to Eq. (1), there is a nested relation between these five user groups. For example, the user group with $\alpha=0.4$ contains the users in the user group with $\alpha=0.2$. This definition is problematic and will result in a redundancy in policy optimization for different user groups.
>
> Thank you for your insightful comments. We acknowledge the issue of overlap in our user grouping strategy, but there are two key reasons behind our chosen approach:
>
> 1. Focus on Low-Activity Users: As discussed in the introduction, we can directly exploit the well-established preferences of high-activity users more effectively due to the richer data available. Conversely, low-activity users stand to gain more from high-intensity exploration, as it helps uncover their latent interests. This observation is validated by our validation experiments in the introduction section. In our method, we specifically target the lower activity users by grouping them based on the bottom quantiles of the return distribution. This approach ensures that our system consistently prioritizes the exploration of interests among these users across different actors in the model. As a result, no matter which actor is served during the inference phase, the interests of low-activity users are always taken into account. By targeting the bottom quantiles of the return distribution, we ensure a concentrated focus on low-activity users, thereby capturing their interests more effectively.
>
> 2. Overestimation of Q-Values: Q-value overestimation in Reinforcement Learning [1-5] refers to the tendency of Q-learning algorithms to overestimate the value of actions, particularly in environments with noise and uncertainty. This can lead to suboptimal policies and erratic learning behavior. In our case, when users are grouped into multiple non-overlapping groups, this issue intensifies due to the challenge of accurately evaluating Q-values in more narrowly defined user groups. Such overestimation leads to incorrect assessments of the Q-value, which can impede convergence, destabilize learning, and reduce the efficiency of exploration.
>
> To empirically validate our approach, we conducted an analysis comparing the performance of our original version against a non-overlapping version in our updated version of manuscript. The results, presented in **Table 8 of Appendix G.3** and below, show that our version performs slightly better than the non-overlapping version. Additionally, in **Figure 8 of Appendix G.3**, we illustrate the Q-value distributions for both versions. This figure highlights the more noticeable overestimation in the Q-values for the non-overlapping version.
>
> |  | KuaiRand/Total Reward | KuaiRand/Depth | ML1M/Total Reward | ML1M/Depth | RL4RS/Total Reward | RL4RS/Depth |
> | --- | --- | --- | --- | --- | --- | --- |
> | UOEP(w/o overlap) | 14.37±0.26 | 14.97±0.24 | 16.60±0.06 | 16.96±0.52 | 8.24±0.31 | 9.43±0.28 |
> | UOEP(ours) | 14.39±0.10  | 14.98±0.09  | 16.59±0.11  | 16.95±0.10  | 8.48±0.53  | 9.60±0.50 |
>
>
> Therefore, while a non-overlapping grouping might be more direct, we opted for our current approach due to these reasons. This strategy offers a balanced focus on low-activity users and mitigates the challenges associated with Q-value overestimation, ensuring more effective and stable exploration policies.
>
> [1] Li Z, Hou X. Mixing update q-value for deep reinforcement learning[C]//2019 International Joint Conference on Neural Networks (IJCNN). IEEE, 2019: 1-6.
>
> [2] Van Hasselt H, Guez A, Silver D. Deep reinforcement learning with double q-learning[C]//Proceedings of the AAAI conference on artificial intelligence. 2016, 30(1).
>
> [3] Prabhakar P, Yuan Y, Yang G, et al. Multi-objective Optimization of Notifications Using Offline Reinforcement Learning[C]//Proceedings of the 28th ACM SIGKDD Conference on Knowledge Discovery and Data Mining. 2022: 3752-3760.
>
> [4] Duan J, Guan Y, Li S E, et al. Distributional soft actor-critic: Off-policy reinforcement learning for addressing value estimation errors[J]. IEEE transactions on neural networks and learning systems, 2021, 33(11): 6584-6598.
>
> [5] Kuznetsov A, Shvechikov P, Grishin A, et al. Controlling overestimation bias with truncated mixture of continuous distributional quantile critics[C]//International Conference on Machine Learning. PMLR, 2020: 5556-5566.

---

> ### Author Response · Authors · 2023-11-15
> **Response to Reviewer 3uff, Part 2**
>
> > W2. The motivation of this paper is to design a separate user-oriented exploration policy for different user groups. However, to my understanding, there is no explicit exploration strategy design for different user groups, and only the optimization objective $\operatorname{CVaR}_\alpha$ varies for different user groups, which does not necessarily promote user exploration for different groups.
>
> We appreciate your observations concerning the exploration strategies for different user groups in our paper. The user grouping by quantiles of return distribution in our UOEP indeed serves as a mechanism to customize the intensity of exploration to the varying levels of user activity.
>
> While the optimization objective $\operatorname{CVaR}_\alpha$ is different across user groups, it is not the sole factor promoting exploration for each group. We can use tailored exploration strategies in different user groups, such as traditional epsilon-greedy strategy. In our experiment, the epsilon values were carefully selected and fine-tuned to reflect the distinct interaction patterns and activity levels of different user groups.
>
> For instance, a smaller epsilon for highly active users aligns with their needs and utilizes their rich interaction history for more precise exploitation. In contrast, a higher epsilon for less active users encourages the introduction of diverse recommendations, compensating for their limited interaction data and potentially enhancing user engagement by exposing them to new content.
>
> Beyond epsilon adjustments and different learning objectives (CVaRs), our introduction of a population diversity loss function is a deliberate and explicit strategy to enhance user exploration. This function incentivizes diverse exploration behaviors across actors, with each actor exploring different parts of the action space in ways that are specifically tailored to the interests and activities of their designated user groups.
>
> The efficacy of this design is substantiated "How does UOEP work?" in Section 4.3, where we demonstrate UOEP's capability to navigate and exploit the action space effectively for different user groups. We regret any confusion caused by our draft. In the revised manuscript, we will provide a detailed explanation of our strategies and more explicitly demonstrate their effects on user exploration for different levels of user activity.

---

> ### Author Response · Authors · 2023-11-15
> **Response to Reviewer 3uff, Part 3**
>
> > W3. The used evaluation metrics (total reward and Depth) do not validate the effectiveness of exploration. Other exploration-related evaluation metrics such as diversity and coverage are necessary to demonstrate the exploration performance.
>
> Thank you for pointing out the importance of using diverse metrics to evaluate exploration effectiveness. Our selection of total reward and depth as primary metrics was driven by our focus on enhancing the long-term performance of our recommendation system, rather than solely on diversifying recommendations or addressing the cold-start problem. This is in line with established practices in RL and RL-based recommendation system literature, where exploration is often employed to optimize for long-term returns.
>
> For example, in diversity-based RL method [1-4], the purpose of exploration is to improve the final return. Similarly, RL-based recommendation systems that prioritize exploration, like in [5-8], concentrate on improving users' long-term engagement. Specifically, Large-scale online experiments conducted by Google [8] demonstrate that user retention and activity levels can be positively impacted through extensive exploration. These findings highlight the potential power of utilizing exploration capability of RL methods in recommendation systems to improve users' experience in the long term. Our selection of baselines, therefore, stems from the realm of RL methods, including conventional RL methods and exploration-oriented RL methods. This sets our approach apart from other studies that focus primarily on diversity or addressing cold-start issues.
>
> Due to constraints in accessing online environments, the direct impact of our method on user engagement over the long term in the real world remains unobserved. However, our offline simulation experiments provide a comparative analysis of various RL-based methods, substantiating the effectiveness of our approach in this context.
>
> However, acknowledging your concern, we have expanded our experimental analysis to encompass additional aspects, specifically focusing on diversity and coverage. To this end, we selected coverage and Intra-List Similarity (ILS) [9] as key metrics for evaluating the effectiveness of our method compared to other exploration-oriented baselines. The accompanying table illustrates that our method notably surpasses these baselines in performance, with the exception of a second-best performance of ILS in the RL4RS dataset. This result underscores the efficacy of the exploration strategy developed in our research, demonstrating its superiority in enhancing both the diversity and coverage of recommendations.
>
> |  | KuaiRand/Coverage | KuaiRand/ILS | ML1M/Coverage | ML1M/ILS | RL4RS/Coverage | RL4RS/ILS |
> | --- | --- | --- | --- | --- | --- | --- |
> | Wolpertinger | 0.00076 | 0.60 | 0.00078 | 0.49 | 0.00078 | 0.47 |
> | HAC | 0.00082 | 0.65 | 0.00081 | 0.30 | 0.00082 | 0.59 |
> | UOEP | 0.00122 | 0.48 | 0.00102 | 0.24 | 0.00108 | 0.52 |
>
> Furthermore, as elaborated in Section 4.3, titled 'How does UOEP work?', our algorithm showcases extensive coverage from a lateral perspective by serving diverse policies across the action space, resulting in a broader range of recommendation outcomes. We are confident that this illustrates the efficacy of our exploration strategy.
>
> We will ensure that our revised manuscript reflects these considerations and provides a balanced view of our method's performance across a range of relevant metrics.
>
> [1] Parker-Holder J, et al. Effective diversity in population-based reinforcement learning. Advances in Neural Information Processing Systems, 2020.
>
> [2] Cideron G, et al. Qd-rl: Efficient mixing of quality and diversity in reinforcement learning. arXiv preprint arXiv:2006.08505, 2020.
>
> [3] Eysenbach B, et al. Diversity is All You Need: Learning Skills without a Reward Function. International Conference on Learning Representations, 2018.
>
> [4] Hartikainen K, et al. Dynamical Distance Learning for Semi-Supervised and Unsupervised Skill Discovery. International Conference on Learning Representations, 2019.
>
> [5] Liu S, et al. Exploration and Regularization of the Latent Action Space in Recommendation. Proceedings of the ACM Web Conference 2023, 2023.
>
> [6] Zhu Z, Van Roy B. Deep exploration for recommendation systems. Proceedings of the 17th ACM Conference on Recommender Systems, 2023.
>
> [7] Yan S, et al. Teach and Explore: A Multiplex Information-guided Effective and Efficient Reinforcement Learning for Sequential Recommendation. ACM Transactions on Information Systems, 2023.
>
> [8] Chen M, et al. Values of user exploration in recommender systems. Proceedings of the 15th ACM Conference on Recommender Systems, 2021.
>
> [9] Ziegler C N, et al. Improving recommendation lists through topic diversification. Proceedings of the 14th international conference on World Wide Web, 2005.

---

> ### Author Response · Authors · 2023-11-20
> **Dear reviewer, do we address your questions?**
>
> Dear reviewer,
>
> We would like to ask whether we have addressed your questions. Please let us know if our reply has addressed your concerns or if any other clarifications are needed. Thanks a lot.

---

> ### Author Response · Authors · 2023-11-22
> **Replying to Reviewer 3uff**
>
> Dear Reviewer 3uff,
>
> As the discussion period is approaching the end, we again extend our gratitude for your reviews that help us improve our paper. Any further comments on our rebuttal or the revised paper are appreciated. Thanks for your time!

---

> ### Author Response · Authors · 2023-11-23
> **Replying to Reviewer 3uff**
>
> Dear Reviewer 3uff,
>
> As the discussion period is approaching the end, we again extend our gratitude for your reviews that help us improve our paper. Any further comments on our rebuttal or the revised paper are appreciated. Thanks for your time!

---

### Official Review · Reviewer_L2ep · 2023-11-01

**Soundness:** 4 excellent
**Presentation:** 4 excellent
**Contribution:** 4 excellent
**Rating:** 8
**Confidence:** 3

**Summary:**

This paper proposed a user-oriented exploration policy approach to facilitate fine-grained exploration with respect to user different activity levels. Specifically, it consists of a distributional critic that allows optimization at different quantiles; and a population of actors optimizing towards different return distributions. With several regularization losses to control diversity and stability, it demonstrates the superior performance with the proposal approaches by comparing to several baselines on public datasets.

**Strengths:**

1. The paper studies an important problem in recommendation system, optimizing user experience with respect to different activity level.
2. The paper is well motivated, and demonstrates to be a superior approach with several baselines and datasets.
3. The paper is clearly written and easy to follow.

**Weaknesses:**

The proposed approach is similar to an ensemble approach in inference. in the real world, such policy might encounter much more expensive serving cost with millions and even billions of action space, which might prevent itself from its adoption.

Also listed several questions down below.

**Questions:**

1. How does different quantile correspond to different exploration strengths?
2. Usually, activity levels are defined by the total volume of user engagement (clicks), instead of ctr. So it's possible that users have very few impressions, but high ctr. In that case, these users are still referred to as low-activity users. How does that affect the results?
3. In section 4.4, the paper only reported the superior performance for low-activity users only. However, it would also be good to report that for high-activity users as well.

---

> ### Author Response · Authors · 2023-11-15
> **Response to Reviewer L2ep, Part 1**
>
> We appreciate your detailed and valuable feedback on our paper.
> > W1. The proposed approach is similar to an ensemble approach in inference. in the real world, such policy might encounter much more expensive serving cost with millions and even billions of action space, which might prevent itself from its adoption.
>
> Thank you for highlighting the potential increase in computational costs associated with our approach when applied to a real-world setting with an extensive action space. We recognize that deploying such a system at scale could entail higher serving costs.
>
> To address this, our framework is designed with flexibility. While SASRec serves as the backbone in our study, the architecture allows for the integration of more efficient backbones suited for industrial applications. The use of shared lower layers across multiple actors, with differentiated MLP networks attached only at the top, effectively reduces the overall parameter count.
>
> Moreover, we propose the deployment of our model specifically during the re-ranking or final ranking phase, where the number of candidate actions is drastically lower than the initial retrieval stage. This stage's action space is more manageable and is indeed comparable to the scope of our experimental setup. This adaptability, coupled with the designed efficiency of our model, mitigates the impact of large action spaces and makes the approach more feasible for practical use.
>
> > Q1. How does different quantile correspond to different exploration strengths?
>
> Thank you for your question regarding the relationship between different quantiles and exploration strengths in our UOEP algorithm. Grouping users by quantiles of return distribution is to tailor exploration intensity to user activity levels. Our empirical findings suggest that a one-size-fits-all exploration strategy is suboptimal due to the diverse range of user activity.
>
> However, when users are grouped based on their activity levels, applying a traditional exploration strategy such as epsilon-greedy can yield notable performance. For example, a smaller epsilon is more suitable for highly active users, ensuring a stable experience with less random exploration, whereas a larger epsilon benefits less active users by introducing more new results into their recommendations. The selection of epsilon values can be fine-tuned according to empirical observations and adjustments reflecting real-world user engagement patterns.
>
> In our UOEP method, we enhance exploration by introducing a population diversity loss. This loss function encourages a varied exploration behavior among actors, ensuring that each one explores distinct aspects of the action space, catering to the preferences of different user groups. Section 4.3 "How does UOEP work?" of our paper provides empirical evidence of how effectively UOEP navigates the action space, achieving a broad and customized exploration for each user group.
>
> > Q2. Usually, activity levels are defined by the total volume of user engagement (clicks), instead of ctr. So it's possible that users have very few impressions, but high ctr. In that case, these users are still referred to as low-activity users. How does that affect the results?
>
> Thank you for your valuable suggestion. We recognize that CTR may not always be a reliable indicator of a user's overall activity level, particularly in scenarios like the one you described where a user has a high CTR but a limited number of impressions. This realization is indeed crucial, as it could potentially impact the validity of our validation experiments.
>
> In light of this, we conducted a detailed analysis of the relationship between CTR and total click count in our KuaiRand dataset. This analysis, which can be found in **Appendix G.1** of the revised manuscript, indicates that users with fewer total clicks generally have lower CTR values. Moreover, we observed a trend where users' CTR tends to increase as their total number of clicks rises. This suggests that the scenario of users with few impressions but high CTR, while possible, is not common in our dataset and thus does not substantially impact our conclusions.
>
> In response to your specific concern, we have also re-executed our validation experiments, this time using the total number of clicks as the basis for grouping users. The results of these experiments, also included in **Appendix G.1**, show a consistent correlation between users' activity levels and their preference for noise, in line with our initial findings. This further analysis corroborates our original conclusions, affirming that our observations are consistent across different activity level metrics.

---

> ### Author Response · Authors · 2023-11-15
> **Response to Reviewer L2ep, Part 2**
>
> > Q3. In section 4.4, the paper only reported the superior performance for low-activity users only. However, it would also be good to report that for high-activity users as well.
>
> Thank you for your valuable suggestion. In response, we have conducted additional experiments to assess the performance of our method with high-activity users. The results are now included in the revised manuscript, specifically under a new table. In this table, we present the average values of the Total Reward distribution's upper 40\% and 50\%, denoted as $\operatorname{ATR} _ {0.4}$ and $\operatorname{ATR}_{0.5}$, respectively, across the test set.
>
> It's important to note that the maximum possible return is capped at 20, corresponding to our experimental setup of the maximum number of user-item interactions per session. The updated results indicate that our method outperforms all baselines across three datasets, with the exception of a second-best performance in the RL4RS dataset.
>
> However, the extent of improvement for high-activity users, as observed in these additional results, may not be as noticeable as the results presented in section 4.4. This is because our method primarily focuses on exploration, which tends to benefit low-activity users to a greater extent. Thus, while our approach shows effectiveness across the user activity spectrum, its impact is more significant among users with lower activity levels.
>
> |  | KuaiRand/$\operatorname{ATR}_{0.4}$ | KuaiRand/$\operatorname{ATR}_{0.5}$ | ML1M/$\operatorname{ATR}_{0.4}$ | ML1M/$\operatorname{ATR}_{0.5}$ | RL4RS/$\operatorname{ATR}_{0.4}$ | RL4RS/$\operatorname{ATR}_{0.5}$ |
> | --- | --- | --- | --- | --- | --- | --- |
> | SL | 20.00 | 20.00 | 19.91 | 19.75 | 10.35 | 10.13 |
> | A2C | 18.94 | 17.83 | 18.46 | 17.94 | 11.06 | 10.85 |
> | DDPG | 19.14 | 18.29 | 19.99 | 19.90 | 10.14 | 10.05 |
> | TD3 | 19.70 | 19.16 | 19.86 | 19.77 | 9.66 | 9.44 |
> | Wolpertinger | 20.00 | 19.96 | 20.00 | 19.99 | 10.50 | 10.29 |
> | HAC | 20.00 | 19.89 | 20.00 | 19.99 | 9.91 | 9.80 |
> | UOEP | 20.00 | 20.00 | 20.00 | 20.00 | 10.85 | 10.78 |

---

### Official Review · Reviewer_BY7j · 2023-11-06

**Soundness:** 3 good
**Presentation:** 3 good
**Contribution:** 2 fair
**Rating:** 5
**Confidence:** 3

**Summary:**

The authors proposed the UOEP framework, which is an RL based recommendation system that can customize the exploration intensity for different user activity levels. Specifically, the authors define the activity level of users based on the return distribution under different quantiles and the framework learns multiple actors where each actor corresponds to a specific user group with a predefined level of activity. The authors conduct extensive offline analysis based on 3 public datasets KuaiRand1K, ML1M and RL4RS.

**Strengths:**

Strength

- The idea of providing different exploration intensity for different user cohorts is very practical and intuitively making sense.

- Proposed algorithms outperform the baselines in various offline analyses. Source code is provided and datasets are public, which make the results easier to reproduce.

- The paper is in general well written

**Weaknesses:**

I will combine both of my concerns and questions with this paper in this section.

1. Although the user argues "user behaviors with different activity levels require varying intensity of exploration, while previous studies often overlook this aspect and apply a uniform exploration strategy to all users", this is not true.

Exploration for different user activity levels (especially designing exploration strategies for new/cold-start/low-engagement users) are very common projects for industrial recommenders with a lot of existing strategies. In the domain of active learning, there are also a lot of previous works that proposed similar ideas to conduct user-side active learning based on criteria like activity level, popularity, prediction uncertainty etc. These existing works make the core technical contribution of this paper become more incremental.

2. In this paper, "the framework essentially learns multiple actors where each actor predicts for a specific user group with a predefined level of activity", this essentially leads to an increase of effective model size(multiple-actors instead of single actor). How much of the gain comes from a larger model size and how much is coming from a more effective exploration strategy?


3. In the introduction session, the quantile of CTR was used to illustrate the user's activity level. Is this reasonable? For example, in an extreme case, a new user with 1 impression and 1 click will lead to a 100% CTR but the system still knows little about him and needs more intensive exploration. Shouldn't metrics like total number of clicks etc be more suitable in this case?

**Questions:**

Please refer to the section above

---

> ### Author Response · Authors · 2023-11-15
> **Response to Reviewer BY7j, Part 1**
>
> Thank you for your valuable feedback on our paper.
> > W1. Although the user argues "user behaviors with different activity levels require varying intensity of exploration, while previous studies often overlook this aspect and apply a uniform exploration strategy to all users", this is not true.
>
> Thank you for your insightful comments. We acknowledge the breadth of work in designing exploration strategies for users at varying levels of activity, particularly in cold-start and active learning domains. Indeed, projects catering to new, cold-start, or low-engagement users are well-documented, employing strategies based on activity level, popularity, and prediction uncertainty. For instance:
>
> Cold start:
>
> + Zhang X, Kuang Z, Zhang Z, et al. Cold & Warm Net: Addressing Cold-Start Users in Recommender Systems[C]//International Conference on Database Systems for Advanced Applications. Cham: Springer Nature Switzerland, 2023: 532-543.
> + Briand L, Salha-Galvan G, Bendada W, et al. A semi-personalized system for user cold start recommendation on music streaming apps[C]//Proceedings of the 27th ACM SIGKDD Conference on Knowledge Discovery & Data Mining. 2021: 2601-2609.
>
> Active Learning:
> + Bu Y, Small K. Active learning in recommendation systems with multi-level user preferences[J]. arXiv preprint arXiv:1811.12591, 2018.
> + Karimi R, Freudenthaler C, Nanopoulos A, et al. Active learning for aspect model in recommender systems[C]//2011 IEEE Symposium on Computational Intelligence and Data Mining (CIDM). IEEE, 2011: 162-167.
>
> However, our paper's use of the term "exploration" is specifically framed within reinforcement learning-based recommendation systems (RL-based RS). This approach significantly differs from the methods employed in cold-start and active learning domains.
>
> In RL-based RS, the primary focus is on enhancing users' long-term engagement. This is achieved through an interactive learning paradigm that dynamically adjusts to user feedback. Unlike traditional recommendation systems that may prioritize immediate user preferences, RL-based RS are designed to discover and cater to evolving user interests over time. This involves a strategic balance between exploration and exploitation, where the system not only leverages existing user data to provide relevant recommendations (exploitation) but also ventures into less-charted territories to uncover potential interests (exploration).
>
> However, the above methods on active learning or cold-start problems can not directly fit in this interactive paradigm. The novelty of our work lies in adapting the exploration at user the level in an RL context, which differs significantly from the typical application of user exploration strategies in cold starts or active learning works. Meanwhile, existing RL-based RS methods [1-4] have not considered exploration at the user level, instead using a relatively data-independent exploration strategy or simply use cold start metric as a reward signal. Thus, our approach considers user-oriented exploration strategies, which is a less explored dimension in existing RL-based RS literature.
>
> We recognize that our draft may not have fully captured the distinctiveness of our contribution, which may have led to misunderstanding. To clarify, our contribution differs from the scope of cold start and active learning by integrating user activity levels into the exploration and exploitation framework of RL. We believe this to be a meaningful step forward in recommendation areas.
> We will revise our manuscript to more clearly define the scope and contribution of our work in the context of existing literature. We will also refine our discussion in the related work section to accurately reflect how our proposed method diverges from and complements existing works.
>
> [1] Liu S, Cai Q, Sun B, et al. Exploration and Regularization of the Latent Action Space in Recommendation[C]//Proceedings of the ACM Web Conference 2023. 2023: 833-844.
>
> [2] Zhu Z, Van Roy B. Deep exploration for recommendation systems[C]//Proceedings of the 17th ACM Conference on Recommender Systems. 2023: 963-970.
>
> [3] Chen M, Beutel A, Covington P, et al. Top-k off-policy correction for a REINFORCE recommender system[C]//Proceedings of the Twelfth ACM International Conference on Web Search and Data Mining. 2019: 456-464.
>
> [4] Chen M, Wang Y, Xu C, et al. Values of user exploration in recommender systems[C]//Proceedings of the 15th ACM Conference on Recommender Systems. 2021: 85-95.

---

> ### Author Response · Authors · 2023-11-15
> **Response to Reviewer BY7j, Part 2**
>
> > W2. In this paper, "the framework essentially learns multiple actors where each actor predicts for a specific user group with a predefined level of activity", this essentially leads to an increase of effective model size(multiple-actors instead of single actor). How much of the gain comes from a larger model size and how much is coming from a more effective exploration strategy?
>
> We appreciate your inquiry into the relative contributions of model size and exploration strategy to our framework's performance. We have indeed delved into this topic in our ablation study, specifically in the experiment titled "Effect of the Distributional Critic" in Section 4.3 of our paper.
>
> In this study, the UOEP variant with a deterministic critic indeed has a larger parameters due to multiple actors. Yet, as illustrated in **Table 2 of Section 4.3**, without employing our exploration strategy within each user group, this variant did not outperform the HAC baseline, which utilizes the smaller number of parameters (both HAC and our method use SASRec as backbone). This outcome indicates that merely increasing model size does not translate to better performance, echoing the difficulty of achieving convergence with larger models in reinforcement learning.
>
> Conversely, our proposed UOEP method, which incorporates a tailored exploration strategy, demonstrates superior long-term performance over both the deterministic critic variant of UOEP and the HAC baseline, as shown in **Table 1 of Section 4.2**. This significant enhancement suggests that the gains are not solely due to an increase in model size but are attributed mainly to the efficacy of the exploration strategy.
>
> > W3. In the introduction session, the quantile of CTR was used to illustrate the user's activity level. Is this reasonable? For example, in an extreme case, a new user with 1 impression and 1 click will lead to a 100% CTR but the system still knows little about him and needs more intensive exploration. Shouldn't metrics like total number of clicks etc be more suitable in this case?
>
> Thank you for your insightful suggestion. We acknowledge that a high CTR may not always accurately reflect a user's activity level, potentially impacting the reliability of our validation experiment. To address this, we analyzed the relationship between CTR and the total number of clicks in our KuaiRand dataset, as detailed in **Figure 6 of Appendix G.1** of the revised manuscript. This analysis reveals that users with a lower total click count typically also exhibit lower CTR values on average. Additionally, it is observed that as the total number of clicks increases, there is a corresponding upward trend in users' CTR. This finding suggests that the extreme case you mentioned, while possible, is relatively rare in our dataset and does not significantly affect our conclusions.
>
> Furthermore, to address your concerns more directly, we have re-conducted the validation experiments using the total number of clicks as a grouping metric. The results, also presented in **Figure 7 of Appendix G.1**, demonstrate a trend between users' activity levels and their preference for noise that is consistent with our previous findings. This additional analysis reinforces our original conclusions, showing that our observations hold true across different metrics.

---

> ### Author Response · Authors · 2023-11-20
> **Dear reviewer, do we address your questions?**
>
> Dear reviewer,
>
> We would like to ask whether we have addressed your questions. Please let us know if our reply has addressed your concerns or if any other clarifications are needed. Thanks a lot.

---

> ### Author Response · Authors · 2023-11-22
> **Replying to Reviewer BY7j**
>
> Dear Reviewer BY7j,
>
> As the discussion period is approaching the end, we again extend our gratitude for your reviews that help us improve our paper. Any further comments on our rebuttal or the revised paper are appreciated. Thanks for your time!

---

> ### Author Response · Authors · 2023-11-23
> **Replying to Reviewer BY7j**
>
> Dear Reviewer BY7j,
>
> As the discussion period is approaching the end, we again extend our gratitude for your reviews that help us improve our paper. Any further comments on our rebuttal or the revised paper are appreciated. Thanks for your time!

---

### Author Response · Authors · 2023-11-15
**To all Reviewers**

Dear Reviewers,

Thank you for your insightful feedback and valuable suggestions. In response to your comments, we have made comprehensive revisions to our manuscript. The updated version of the paper now includes detailed responses to the majority of the concerns raised. We have addressed these points in **Appendices G and H**, which have been carefully crafted to provide clarity and additional information.

We believe these enhancements will facilitate a better understanding of our research and its implications. We invite all reviewers to examine these sections in the revised PDF, which we hope will address any remaining questions and further elucidate our study's contributions.

We appreciate the opportunity to improve our work based on your feedback and look forward to any additional comments you may have.

Sincerely,
Submission 647

---

### Meta-Review · Area_Chair_tEg5 · 2023-12-14

**Metareview:**

This paper proposed UOEP: user-oriented exploration policy approach to facilitate fine-grained exploration with respect to different user activity levels. The framework consists of a distributional critic that allows optimization at different quantiles; and a population of actors optimizing towards different return distributions. With several regularization losses to control diversity and stability, it demonstrates the superior performance with the proposal approaches by comparing to several baselines on public datasets with online simulator.

Strength: the idea of providing different exploration intensity for different user cohorts is very practical and intuitively making sense. The empirical results show promise and the ablation studies provides insights on the effect of different components employed.
Weakness: while the paper is motivated designing separate user-oriented exploration policies for different user groups, the main working components of the framework, i.e., distributional critics, multiple actors optimizing different CVaR_\alpha, is not specific to exploration, but more towards the general direction of designing different policies for different user groups, which calls for different baselines to compare such as simply dividing the user groups and optimizing different policies for each.

**Justification For Why Not Higher Score:**

While the paper is motivated designing separate user-oriented exploration policies for different user groups, the main working components of the framework, i.e., distributional critics, multiple actors optimizing different CVaR_\alpha, is not specific to exploration, but more towards the general direction of designing different policies for different user groups, which calls for different baselines to compare such as simply dividing the user groups and optimizing different policies for each. Without such simple baselines, it is hard to assess the merit of the proposed framework.

**Justification For Why Not Lower Score:**

N/A

---

### Decision · Program_Chairs · 2024-01-16

Reject